# Effects of Supplementation of a Mycotoxin Mitigation Feed Additive in Lactating Dairy Cows Fed *Fusarium* Mycotoxin-Contaminated Diet for an Extended Period

**DOI:** 10.3390/toxins15090546

**Published:** 2023-09-04

**Authors:** Alessandro Catellani, Francesca Ghilardelli, Erminio Trevisi, Alessio Cecchinato, Vittoria Bisutti, Francesca Fumagalli, H. V. L. N. Swamy, Yanming Han, Sandra van Kuijk, Antonio Gallo

**Affiliations:** 1Department of Animal Science, Food and Nutrition (DIANA), Università Cattolica del Sacro Cuore, Via Emilia Parmense, 84, 29100 Piacenza, Italy; alessandro.catellani@unicatt.it (A.C.); francesca.ghilardelli@unicatt.it (F.G.); erminio.trevisi@unicatt.it (E.T.); francesca.fumagalli@unicatt.it (F.F.); 2Department of Agronomy, Food, Natural Resources, Animals and Environment (DAFNAE), University of Padova, Viale dell’Università, 16, 35020 Legnaro, Italy; alessio.cecchinato@unipd.it (A.C.); vittoria.bisutti@unipd.it (V.B.); 3Selko Feed Additives, Nutreco, Stationsstraat 77, P.O. Box 299, 3800 AG Amersfoort, The Netherlands; swamy.haladi@selko.com (H.V.L.N.S.); yanming.han@nutreco.com (Y.H.); sandra.van.kuijk@selko.com (S.v.K.)

**Keywords:** mycotoxin, animal, blood parameters, milk quality, animal welfare

## Abstract

*Fusarium* mycotoxins are inactivated by rumen flora; however, a certain amount can pass the rumen and reticulum or be converted into biological active metabolites. Limited scientific evidence is available on the impact and mitigation of *Fusarium* mycotoxins on dairy cows’ performance and health, particularly when cows are exposed for an extended period (more than 2 months). The available information related to these mycotoxin effects on milk cheese-making parameters is also very poor. The objective of this study was to evaluate a commercially available mycotoxin mitigation product (MMP, i.e., TOXO^®^ HP-R, Selko, Tilburg, The Netherlands) in lactating dairy cows fed a *Fusarium* mycotoxin-contaminated diet, and the repercussions on the dry matter intake, milk yield, milk quality, cheese-making traits and health status of cows. The MMP contains smectite clays, yeast cell walls and antioxidants. In the study, 36 lactating Holstein cows were grouped based on the number of days of producing milk, milk yield, body condition score and those randomly assigned to specific treatments. The study ran over 2 periods (March/May–May/July 2022). In each period, six animals/treatment were considered. The experimental periods consisted of 9 days of adaptation and 54 days of exposure. The physical activity, rumination time, daily milk production and milk quality were measured. The cows were fed once daily with the same total mixed ration (TMR) composition. The experimental groups consisted of a control (CTR) diet, with a TMR with low contamination, high moisture corn (HMC), and beet pulp; a mycotoxins (MTX) diet, with a TMR with highly contaminated HMC, and beet pulp; and an MTX diet supplemented with 100 g/cow/day of the mycotoxin mitigation product (MMP). The trial has shown that the use of MMP reduced the mycotoxin’s negative effects on the milk yield and quality (protein, casein and lactose). The MTX diet had a lower milk yield and feed efficiency than the CTR and MMP HP-R diets. The MMP limited the negative effect of mycotoxin contamination on clotting parameters, mitigating the variations on some coagulation properties; however, the MMP inclusion tended to decrease the protein and apparent starch digestibility of the diet. These results provide a better understanding of mycotoxin risk on dairy cows’ performances and milk quality. The inclusion of an MMP product mitigated some negative effects of the *Fusarium* mycotoxin contamination during this trial. The major effects were on the milk yield and quality in both the experimental periods. These results provide better insight on the effects of mycotoxins on the performance and quality of milk, as well as the cheese-making traits. Further analyses should be carried out to evaluate MMP’s outcome on immune–metabolic responses and diet digestibility.

## 1. Introduction

Mycotoxins are a group of metabolites produced by several filamentous fungi in the genera *Aspergillus*, *Fusarium* and *Penicillium* that can induce toxic responses when ingested by humans and other living organisms [1,2,3]. Mycotoxins are stable and can be detected in animal feeds and forages [4,5,6]. *Fusarium* mycotoxins are widespread, and they contaminate field crops in all climatic zones where plant material is available [7]. Previous studies have described the toxicological effects of *Fusarium* mycotoxins in farming animals [2,8,9,10,11]. The administration of deoxynivalenol (DON)-contaminated feed to animals can lead to the exhibition of symptoms such as gastrointestinal disorders, soft feces, diarrhea, immunosuppression and decreased performance [9,12,13]. The biological mechanisms that cause these responses are not well understood, but DON can affect rumen health and damage the permeability of the rumen and/or gut epithelia [14].

Fumonisins (FBs) are mycotoxins that are cytotoxic, hepatotoxic and nephrotoxic, although their mechanism of action is unclear in dairy cows [15,16,17,18]. The animals’ guts absorb FBs [19], which can lead to the alteration of some metabolic pathways. An intestinal immunosuppressive condition involves molecular systems, such as the downregulation of MyD88 and TLRs, which are important players in the first phase of pathogen recognition, ultimately leading to the inflammatory cascade [20,21]. In the inflammatory response, a parameter of particular interest is the circulating leukocytes, which is not yet well-understood in ruminants.

According to [22], some mycotoxins could be inactivated by the rumen flora (e.g., DON is partly converted into the less toxic metabolite, DOM-1, in particular rumen physiological conditions), while others pass through the rumen and reticulum (e.g., FBs) or are converted into metabolites while still preserving their biological activity (zearalenone, ZEN). Some mycotoxins can modify rumen flora due to their antimicrobial activity and negatively affect the immune system of the host animals. An example is patulin, which acts against Gram + and Gram − bacteria and protozoa. In vitro studies have reported that it has a negative effect on the production of volatile fatty acids (VFA) and acetate, and on protein synthesis in the rumen fluid. *Ruminoicoccus albus* and *Methanobrevibacter ruminatium* were significantly inhibited by fusaric acid. Other mycotoxins with antimicrobial activity include mycophenolic acid, roquefortine C, beauvericin and enniatins [6]. Impairment of the rumen microflora by mycotoxins results in a reduced filling of the rumen, less feed conversion and diarrhea; these symptoms are accompanied by decreased milk production and increasing incidences of subclinical mastitis with increased somatic cell counts [2,8].

Currently, there is limited scientific evidence in terms of mycotoxins’ effects on the digestion, physiology and pathophysiology of dairy cows [2,6], especially if exposed for an extended period (e.g., more than 2 months). EFSA ranks DON as an undesirable substance of the agri-food chain [12] and it also causes undesirable effects for dairy cows: loss of appetite, reduced rumination activity and feed intake, upregulation of pro-inflammatory cytokines and immuno-suppression, but only few feeding studies with large ruminants are available, as reported in a recent EFSA scientific opinion on FBs [23]. The adverse effects of FB ingestion involve changes in organ appearance and serum enzymes and biochemistry, as well as a reduction of kidney and liver function. Mycotoxin effects on cheese-making parameters have been poorly investigated [24] when mycotoxin mitigation products (MMP) are added to diets. Consequently, an evaluation of mycotoxins’ economic impact (aflatoxins and mycotoxins originating from *Fusarium)* on ruminant livestock production requires further investigation [25,26,27,28]. Advances in this topic may help crops/feed technicians and dairy farmers in dealing with non-specific health and performance inefficiencies on their farms. Among MMP, some commercially available products contain selected glucose biopolymers and purified ß-glucans, as well as bentonite or yeast cell walls, which are able to bind a wide range of mycotoxins and reinforce the intestinal barrier function and the immune system that are mainly affected by *Fusarium* mycotoxins, such as trichothecenes and fumonisins [28]. These types of MMP contain vitamins, antioxidants, yeast cell walls and bentonite clays. The objective of this study was to determine the performance effects of a commercially available MMP, TOXO^®^ HP-R, in lactating dairy cows fed with *Fusarium* mycotoxin-contaminated diets at different levels of contamination and evaluate the effects on the dry matter intake capacity, rumination, immuno-physiological parameters, milk yield and quality, cheese-making traits and the overall health status of multiparous cows. The MMP tested in our trial contains four components, including bentonites, inactivated yeast cell wall fractions, β-1,3/1,6 glucans and antioxidants. It has four layers of protection. The first layer of protection, bentonite, can adsorb a portion of certain dietary mycotoxins to prevent their adsorption in the small intestine [17]. Mycotoxins that escape from the adsorbents may damage the gut barrier function of enterocytes and, afterwards, translocate from the lumen side into the blood stream. The inactivated yeast cell wall fractions showed the efficacy of strengthening the gut barrier function [6], which helps further reduce mycotoxins’ bioavailability and works as the second layer of protection. A portion of dietary mycotoxins, however, may still be absorbed in the small intestine and lead to immunosuppression, especially for the gut’s innate immunity. Beta-glucans, as the third layer of protection, are recognized by the receptors of macrophages and therefore stimulate the innate immune system [6]. Lastly, most mycotoxins present in the blood stream will eventually end up in the liver for detoxification. Scientific research indicates that a dietary increase of antioxidants is beneficial because the antioxidants enhance the detoxification of ingested mycotoxins into non-toxic metabolites and help to overcome mycotoxin-induced oxidative stress, which explains the presence of antioxidants in TOXO HP-R as the fourth layer of protection. As reported above, one of the aims of this study was to evaluate the mitigation of the MMP product of some negative effects of *Fusarium* mycotoxin contamination through long-term exposure (54 days).

## 2. Results 

Table 1 shows the chemical composition, digestibility and energy evaluations of experimental TMR diets fed to lactating dairy cows, as well as their average mycotoxin concentrations. The DM (% as fed) did not differ among the diets, with the highest value in the CRT group (53.8 ± 2.4% as fed) and the lowest in the MTX group (52.7 ± 2.3% as fed). Similarly, the CP, soluble CP and ash contents resulted in very similar levels among the experimental diets, with average values of 14.7% DM, 5.2% DM and 8.8% DM, respectively. The diets did not differ in fiber fractions and NDF digestibility when evaluated after 24 h of rumen incubation, and the differences among the diets can only be considered numerically. Similarly, the average values of aNDFom, ADF, ADL, starch and sugar were 31.7, 19.8, 3.0, 26.9 and 4.2% DM, respectively. The mean values of neutral detergent insoluble crude protein (NDICP) and acid detergent insoluble crude protein (ADICP) were 3.1 ± 0.5 and 0.9 ± 0.1% DM, respectively, for all the experimental diets.

Very similar values among the diets were also observed for the total digestible nutrients (TDN) and metabolizable energy for lactation at three times the maintenance level (ME3x), and average values of 70.1% and 2.54 Mcal/kg DM were respectively measured or calculated. 

Regarding the regulated mycotoxin contaminations of the experimental diets, the highest value of FB1 in TMR was present in the MTX and MMP groups, which had mean FB1 values of 159.5 ± 60.9 and 163.8 ± 58.1 µg/kg DM, respectively, and mean FB2 values of 75.9 ± 31.9 and 77.9 ± 32.9 µg/kg DM, respectively. The CTR had the lowest FB1 and FB2 concentration values, being 85.3 ± 56.3 and 44.3 ± 30.4 µg/kg DM, respectively. The average values of ZEN and DON were 196.8 ± 75.7 and 1021.7 ± 234.5 µg/kg DM in MTX or 248.5 ± 139.3 and 1009.6 ± 213.5 in the MMP groups, respectively. The lowest ZEN and DON concentrations were in the CTR group, with mean values of 43.2 ± 13.1 and 284.9 ± 91.9 µg/kg DM, respectively. 

Table 2 shows the chemical and fermentative (only silage) characteristics of single feeds. The chemical compositions appeared typical for specific ingredients used in this experiment. Furthermore, no evidence of differences in the chemical composition were observed between the low-contaminated (CTR) and highly contaminated (TRT) HMC and the low-contaminated (CTR) or highly contaminated (TRT) beet pulp. The table below reports the mycotoxin contamination of FB1, FB2, ZEN and DON for each ingredient used in the in vivo trial. The main mycotoxin contamination feeds were HMC TRT, highly contaminated by FB1 (645.3 ± 43.2 µg/kg DM), FB2 (278.3 ± 25.9 µg/kg DM) and DON (1988.1 ± 234.2 µg/kg DM) or beet pulp TRT, for which the main mycotoxin was ZEN (2911.4 ± 121.4 µg/kg DM). All the feed in Table 2 follow the specific European recommended levels for DON, ZEN and FBs contamination levels. Only beet pulp TRT showed higher ZEN contamination, even if this feed is lower than 3 mg/kg. 

Table 3 shows the least squares means and associated standard error of the mean (SEM) for the feeding behavior, body weight, milk yields, feed efficiency and milk quality parameters of all the recruited Holstein cows that were fed a control diet without an adsorbent and mycotoxins (CTR) during each adaptation period. The feeding behavior, milk yields and milk parameters had a *p* value of the models for the groups that were not always significant (*p* > 0.05) during the adaptation period, whereas it was possible to observe a tendency towards significance with regards to body weight (*p* = 0.08). This means that before exposure periods, the cows had similar feeding behavior and milk performances, regardless of their assigned groups. The period effect was significant (*p* < 0.05) for the dry DMI (kg/cow/day and % BW), fat- and protein-corrected milk (FPCM) (kg/cow/day) and energy-corrected milk (ECM) (kg/cow/day) milk yields, fat (%), protein (kg/cow/day), casein (kg/cow/day), lactose (kg/cow/day), MUN (mg/100 mL) and LogSCC (Log10(cells/mL)); this was probably due to differences in days in milk. No differences in the milk composition (i.e., fat, protein, casein, lactose, urea and somatic cell counts) were observed among the groups before the exposure period. 

Table 4 shows the least squares means and associated SEM for the feeding behavior, body weight, milk yields, feed efficiency and milk parameters during the exposure period. The DMI did not differ among the treated groups or periods, resulting in an average equal to 26.02 kg DM/cow/day or 4.03% BW. The high intake of the cows was related to the earlier stage of lactation and high production levels. Similarly, the rumination times did not differ among the groups and experiment periods, resulting in an average equal to 514 min per day. The BCS was evaluated for the early lactation cows used in the experiment and it resulted in an average equal to 3.20, based on a 1 to 5 evaluation scale. The BW did not differ among the periods or groups, with the average equal to 647 kg. 

The milk yield did not differ among the experimental groups, despite numeric differences of more than 1.5 kg/cow/day between the MTX and MMP groups; this was probably related to great intra-group variability (i.e., SEM = 0.27). In particular, the CTR and MMP groups numerically produced more milk than the MTX groups, with a difference of +0.7 and +1.8 kg of milk/cow/day, respectively. Day × treatment interactions were reported for 3.5% of fat-corrected milk and energy-corrected milk (*p* < 0.05). For ECM, a higher average production of +1.8/1.9 kg/cow/day was observed in the MMP group as compared to the MTX group. The numerical differences between the CTR and MTX groups for ECM was +0.6 kg/cow/day. Moreover, we detected an interaction effect D * T for ECM and FCM that could explain a variation of the data throughout the experimental weeks of mycotoxin exposure. 

Regarding the milk parameters and nutrient yields, the MMP was characterized by the highest values for fat and lactose, both in concentration and yield. Increases of approximately 5–6% of these milk nutrients were reported and the differences were more pronounced after 6 to 8 weeks of exposure. Similarly, greater values for the protein and casein yields were observed for CTR and MMP as compared to the MTX groups, and the differences tended to be more evident at the end of the exposure periods (day × treatment interactions < 0.05). No differences were found among the groups for MUN and SCC.

Table 5 shows the main nutrient digestibility. The period effects in the apparent NDF, starch and CP digestibility (i.e., −2.2%, −0.7%, and 2.0%, respectively) were measured. A slight reduction in the apparent CP digestibility was observed among the groups, with the MMP diet showing the lowest values (i.e., 79.32% in MMP vs., on average, 81.73% of CTR and MTX diets; *p* < 0.05). Similar results were reported for the apparent NDF digestibility (*p* = 0.126) and apparent starch digestibility (*p* = 0.060). The pH of the fecal samples was higher in the first than the second period, without differences among the treatments. Regarding the volatilome fecal profile, the majority of the analyzed compounds differed at a *p* < 0.05 between the experimental periods, even when no differences among the treatments were measured, with the only exception of iso-valeric acid that tended (*p* = 0.066) to be higher in MTX than the CTR and MMP diets. Only the numerical differences among the treatments (*p* = 0.836) were reported for the total volatile fatty acids, with the lowest concentration being in CTR (i.e., 224.90 mmol/kg DM) and the highest in MMP (i.e., 260.13 mmol/kg DM). 

Table 6 reports the effects of various diets on the hematochemical parameters. Among the indexes of energy and protein metabolism, BOHB tended to be higher (*p* < 0.10) in MTX with respect to the other diets. In addition, cholesterol and urea were numerically higher in MTX in comparison with the other diets. MTX also showed a lower concentration of transaminases (GOT and GGT; *p* > 0.05) with the respect to CTR and MMP. Overall, the presence of the mycotoxin mitigation product (MMP vs. MTX) resulted in very slight differences at the hematochemical levels, which were not statistically supported. Interestingly, the values of the plasma parameters of MMP were very similar to the CTR group, supporting the effective adsorption of mycotoxins during the transit into the gut. The results of some blood parameters (e.g., MPO and alkaline phosphatase with a significant W*T effect) showed interesting numeric differences among the treatments and deserve future investigation.

Table 7 reports the milk coagulation properties of lactating cows fed one of the three tested diets. The type of treatment did not significantly affect most of the tested milk cheese-making traits; however, numerical differences indicating a lower aptitude of milk produced by MTX to become cheese was observed for several parameters, such as r, K20, a30, a45 and a60. The only significant parameter (*p* < 0.05) was Rct_eq (min), which indicated a longer rennet coagulation time. The values for this parameter, based on recommendations from Bittante [30], can be classified as fast-coagulating samples (Rct_eq < 17 min), samples coagulating at an average rate (17 min < Rct_eq < 22 min), and slowly coagulating samples (Rct > 22 min). Milk samples that did not coagulate by the end of the recording time (30 or 90 min) were classified as non-coagulating (NC). Consequently, the milk of the MMP group is considered average, whereas milk from the MTX group is considered a slowly coagulating sample. The tendency (0.05 < *p* < 0.10) in CFmax and CFp were observed for the MTX vs. CTR and MMP groups, with the worst results reported in the MTX treatments. Furthermore, a trend towards significance (0.05 < *p* < 0.10) was observed for the parameter tmax. No other differences were observed for the CY or REC parameters among the groups.

## 3. Discussion

### 3.1. Contamination of Animal Diets

The mycotoxin contamination levels found within the TMRs used in this study are similar to those reported in [31] and well below the EU regulated limits for animal feed when the regulation was declared. The CTR diet presents low contamination levels for all the mycotoxins under investigation, specifically DON (<650 μg/kg), ZEN (<107 μg/kg) and FB (<280 μg/kg). In contrast, the MTX and MMP diets showed high contamination levels for DON (<1000 μg/kg), medium for ZEN (<260 μg/kg) and low for FB (<280 μg/kg). This is quite in line with the distribution of *Fusarium* mycotoxins reported by [31], in that 30% and 33% of the feed samples had low and high DON contamination levels, respectively, while 43% and 33% of the samples had low and medium ZEN contamination levels. Although the authors of [31] reported that 44% of the samples had FB contamination at a medium level, a low contamination level of 20% was found in this trial. The lack of homogeneity between the different batches of mycotoxin-contaminated culture media could be the cause of the higher levels of ZEN contamination found in the MMP diets compared to the MTX diets [31].

### 3.2. Feed Intake and Rumen Activities

Regarding DMI, no significant differences were observed between the groups in this trial; however, compared to the CTR group, the DMI value was numerically slightly higher in MTX and MMP, with the highest values achieved in the MMP diet. Other authors [31,32,33,34] observed that cows maintained a similar DMI despite being fed diets with different mycotoxin levels; however, the study carried out by the authors of [35] observed an increase in DMI following the ingestion of mycotoxin-contaminated diets. This could be a good explanation for higher DMI in diets with a higher level of mycotoxin contamination. The authors of [36] also found changes in DMI following mycotoxin ingestion, such as FB and ZEN, while EFSA [12] reported that DON is associated with a loss of appetite resulting in reduced feed intake. 

Regarding the rumination time, no significant differences were observed; however, the values show that the animals who were fed the MTX diet had numerically shorter rumination times, whereas the animals fed the MMP diet had longer ones. The authors of [13] reported that DON is also responsible for a reduction in ruminal activity, as found in this trial: the MTX diet reduced ruminal activity, as reported in Table 4. Despite high DON contamination, cows who were fed the MMP diet had the longest rumination time. This difference may be due to the presence of the MMP, which, in this case, consists of antioxidants and gut modifiers in addition to bentonite and an immune modulator. Ingestion of mycotoxin-contaminated feed causes oxidative stress [37], and antioxidants in diets help reduce their negative impact [13,38]. Antioxidants, with their beneficial effects, could lead to an increase in the overall health status of cows, thereby promoting an increase in ruminal activity. 

Another trial conducted by [31] observed a completely different trend in the rumination time: the minimum value was found for the mycotoxin-contaminated diet containing a mycotoxin degradation product, while the maximum one occurred in animals fed only the mycotoxin-contaminated diet. The differences could be due to the different composition of the mycotoxin mitigation product or the different dietary level of the mycotoxins. The mechanisms that cause differences in rumination time are not yet well understood and defined; therefore, further experiments will be needed to clarify these aspects.

### 3.3. Milk Yield and Its Quality

The production and milk quality parameters show that there were no significant differences between the three diets tested in this trial. The milk production levels appear to be in line with what was observed by [33], while [32] reported that the administration of mycotoxin-contaminated diets did not affect milk production levels or its composition. The MTX diet led to lower milk yields and worsened quality parameters (except for the fat content, which was higher than that of the CTR diet). The authors of [32] also observed a numerical reduction in milk production in the mycotoxin-exposed group, while [31] noted significantly reduced milk production. These differences represent a serious economic loss at the farm level [31]. The MMP diet resulted in higher milk production levels than the CTR diet, with improved quality traits (except for protein and casein). Similar to what was observed in this trial, the authors of [39,40] observed that the production level increased, while [40] reported that the administration of zeolite as an MMP resulted in a significant reduction in milk fat and protein. In the trial conducted by [31], the diet containing an MMP resulted in higher milk production compared to the contaminated diet; however, the production value of the control animals was higher than all the other diets used in the trial. In contrast, the authors of [24] observed that the use of MMP caused no changes in the milk yield or quality characteristics. 

Milk coagulation parameters have an essential role in the Italian dairy sector, as most of the milk produced is processed into high-quality products [41,42]. No differences were observed for most of the analyzed milk characteristics; the Rct_eq was the only significant result, and CFmax and CFp tended to be significant, with a tendency of an interaction effect W*T. Nevertheless, differences were observed that indicate a lower cheese aptitude of the MTX diet, which is in line with these results. The authors of [43] observed that mycotoxin intake negatively influenced milk cheesemaking, particularly the cheese curd firmness and whey volume. Although not significant, the rennet coagulation time was also influenced by the intake of mycotoxin-contaminated diets. The study conducted by [31] also showed that mycotoxins cause negative effects on milk coagulation, particularly in K20 and a30 parameters, whereas the intake of MMP restores the levels similar to those of the control diet. Additionally, in this study, the MMP diet accelerated the coagulation times. Further studies on the effects of mycotoxins on milk coagulation properties will be necessary to extend the current knowledge. Further studies should be performed to verify the effects of *Fusarium* mycotoxin on milk production with a larger number of cows to verify the numeric differences. 

### 3.4. Digestibility Traits and Ruminal and Fecal Variables

The trial results show that the apparent CP digestibility was significantly lower in the MMP diet than in the other two diets. The reduction in NDF and starch apparent digestibilities were not significant; however, the reduction in the starch digestibility was more significant. The results obtained in the present trial differ with the bibliography: in particular, the authors of [31,44] observed that mycotoxin administration reduced the NDF digestibility, while the authors of [38,45] noted a decrease in CP digestibility following DON- and FB-contaminated diet administration. Further studies are needed to better understand these mechanisms and to identify why the apparent digestibility values were lower in the MMP diet than in the MTX diet.

### 3.5. Immuno-Metabolic Parameters

Regarding the immuno-metabolic parameters, the mycotoxin-contaminated diets caused minimal effects that were not statistically significant, partly in line with the trial conducted by [31]. Only one significant difference was observed in this trial, namely, the BOHB parameter, which increased significantly in the MTX diet compared to the other two diets, differing from what is found in the literature. The authors of [46] observed that in cows fed a mycotoxin-contaminated diet with 60% concentrate, the BOHB value was lower than in the control diet. In this trial, differences could be observed for urea and globulin, and numerical decreases were noted for transaminases (GOT and GGT) and paraoxonase; however, the trend in transaminases differs from that observed by several authors [6,31,47], who found elevated transaminase levels. There is a potential positive impact of product supplementation that can be related to its impact on immunity. Indeed, the groups that received mycotoxins showed a slight increase of MPO, which suggests increased activity of neutrophils and/or monocytes, with a significant interaction W*T. Nevertheless, none of the analyzed immune indexes (i.e., acute phase proteins) or oxidative stress indicators (i.e., ROM) were altered in these cows; thus, all these aspects deserve deeper investigation in the future to be fully elucidated. The administration of the MMP diet did not cause significant differences in the immunological parameters; on the contrary, the blood parameters were very similar to those of the CTR group, supporting the effective reduction of mycotoxin absorption from the gut. Lastly, future trials should be carried out to produce more data to clarify some of the variations in the blood parameters in cows with a long mycotoxin exposure. 

## 4. Conclusions

Dairy cows fed with a diet contaminated with *Fusarium* mycotoxins for an extended period (i.e., 2 months) were adversely impacted. The presence of DON, ZEN and FB in the dairy cow diets, even below the United States FDA and European Union guidelines, negatively impacted the milk yield and milk rennet coagulation properties. The MMP reduced the negative effects of the mycotoxin by increasing the ECM and milk quality (mainly protein, casein, lactose and clotting features). Cows fed a mycotoxin-contaminated diet (MTX) had a lower milk yield and feed efficiency than the CTR and MMP diets, even if these differences were only numerically different. Mycotoxin contamination modified the apparent digestibility of starch and protein and some milk coagulation properties, such as Rct_eq, CFmax and CFp parameters; however, the contamination did not severely modify the energy metabolism, liver functionality, innate immune system or oxidative stress.

These results provide better knowledge of the risks of mycotoxin exposure on the performance and quality of milk. Further analyses should be carried out to verify the effects of mycotoxin mitigation products on immune–metabolic responses and diet digestibility; a larger number of cows should be studied to verify the consistency of these results. This study shows that the use of additives for mycotoxin control can be a valuable strategy to prevent undesirable effects in animals and alleviate their adverse effects. The study of organic and inorganic additives has already been extensively studied, but research in this field is still necessary due to the new emerging technologies available on the market and new knowledge in the field of mycotoxins. Future scenarios in this area of research will be related to climate change and immunometabolic parameters of dairy cattle as marker of animal health. Moreover, future studies should be conducted to better investigate the effects of mycotoxin metabolites in biological fluids or a modified mycotoxin form in feeds when particular effects on animals cannot be fully explained with an analysis of regular mycotoxins. 

## 5. Materials and Methods

The study was authorized by Italian health regulations relating to the accommodation and care of animals used for experimental and other scientific purposes (authorization n. 138/2021-PR, issued on 2 February 2021).

### 5.1. Experimental Cows and Diets 

Thirty-six multiparous mid-lactation Holstein cows were involved in the experiment conducted at the CERZOO research and experimental center (San Bonico, Piacenza, Italy). CERZOO is an innovative agro-zootechnical research farm, and it is equipped with facilities designed to study applications to increase production efficiency, animal welfare and sustainability. In the barn, each cow is equipped with sensors to continuously monitor the state of health, weight, feeding behavior, various activities and the quantity and quality of production. The structures provide an automated system of ventilation, and the cows are constantly monitored for feed intake. In addition, in the veal farm, there are automation systems that allow individual control of the consumption of milk and feed. In the next paragraphs, we report the equipment with which CERZOO is provided, which was used in this trial.

The animals were raised in a free stall, provided with individual feeding stations and had free access to water. To ensure drinking water safety, an annual analysis for sodium, chloride, potassium, calcium, Salmonella spp. and Escherichia coli are performed in agreement with Italian regulations. The study was conducted during 2 periods (18 different cows for each period): spring (from March to May) and summer (from May to July). In each period, six animals/group were considered (12 cows for each experimental group total). The cows were grouped based on the days in milk (DIM), parity, milk yield, body condition score (2.75 ± 0.35 parities and 101 ± 37 days DIM at the onset of the first period or 2.99 ± 0.34 parities and 145 ± 61 days DIM at the onset of the second period) and randomly assigned to specific experimental groups. Each experimental period consisted of 9 days of adaptation, followed by 54 days of exposure. Each cow had free access to a feed bin of the RIC feeding system, which monitored their feed intake and behavior. A pedometer and a rumenometer were used to monitor the activity and rumination time, respectively. The daily milk production was measured by an Afimilk system. The cows were milked twice daily at 4.00 a.m. and 4.00 p.m. The cows were weighed after each milking, and the data were averaged daily. 

The animals were randomly allocated, before the start of the study, to one of three experimental groups, with six dairy cows for each treatment in each experimental period. The treatments consisted of the following: (i) CTR diet, TMR with low contamination HMC (high-moisture corn) and beet pulp; (ii) MTX diet, TMR with highly contaminated HMC and beet pulp; and (iii) MMP diet, MTX diet supplemented with about 100 g/cow/day of MMP (i.e., TOXO^®^ HP-R, Selko, Tilburg, The Netherlands). The number of parties for CTR, MTX and MMP were, respectively, 2.83 ± 0.24, 2.95 ± 0.35 and 2.88 ± 0.4. The DIM for CTR, MTX and MMP were, respectively, 124 ± 47, 131 ± 32 and 128 ± 50. The MY for DIM for CTR, MTX and MMP were, respectively, 39.65 ± 4.5, 40.1 ± 3.4 and 39.95 ± 5.0. The BCS for DIM for CTR, MTX and MMP were, respectively, 3.00 ± 0.1, 2.95 ± 0.25 and 3.1 ± 0.36. 

Each total mixed ration (TMR) used in this study had the same composition, and the cows were fed once a day at 08:00 h. Approximately 5% of expected orts were collected individually and weighed daily. The components (Table 1) were mixed in a mixer wagon (Rotomix 5000, Bravo srl, Cuneo, Italy) in this order: corn silage, barley silage, alfalfa hay, soybean meal (44%), dehulled sunflower meal (34%), salts, mineral–vitamin supplements and water. Low- or high-contaminated high-moisture corn (HMC), cracked cornmeal and beet pulp feed ingredients were added, respectively, to low- (i.e., CTR diet) or high- (i.e., MTX or MMP-diets) contaminated diets, and were mixed using Rotomix 5000 mixer wagon (Bravo, Cuneo, Italy), and then fed to the animals. The treatment consisted of 100 g/cow/day of mycotoxin mitigating product (i.e., TOXO^®^ HP-R) directly added to the highly contaminated TMR, according to a procedure described by [18]. The whole TMR with low- or high-contaminated feed ingredients was provided to RIC based on the measurement of the dry matter intake (DMI) on the previous day. During the adaptation period, all the animals received the same CTR diet, which was similar in composition and contained low-contaminated feed ingredients. Representative dietary samples were collected each week and analyzed for nutrient composition and mycotoxins, as described below. Samples of individual ingredients were analyzed at the beginning and at the end of each experimental period, and their chemical compositions and mycotoxin contaminations were determined (Table 2). 

### 5.2. Analysis of Feeds, Diets and Mycotoxins

TMR samples were collected weekly, whereas the feed samples were taken at the start and end of each exposure period. We collected TMR seven times for each experimental period for each cow group (in total, 14 TMR sampling for CTR, 14 for MTX and 14 for the MMP group). The chemical profile (i.e., TMR) and mycotoxin levels were evaluated (i.e., both TMR and low- and high-contaminated feed ingredients), as reported in [31]. Before the analysis, the samples were dried at 60 °C in a ventilated oven for 48 h and milled through a 1 mm screen using a laboratory mill (Thomas-Wiley, Arthur H. Thomas Co., Philadelphia, PA, USA), after which they were stored for subsequent analysis. The dry matter (DM) was determined by the gravimetric loss of free water after heating at 105 °C for 3 h (Association of Official Analytical Chemists [48], (method 945.15)). The DM concentration was calculated after drying the feed and TMR samples at 60 °C in a ventilated oven for 48 hr. The ash was considered gravimetric residue after incineration at 550 °C for 2 h [48] (method 942.05). An ether extract (EE) was obtained using the method proposed by [48], method 920.29, and the crude protein (CP; N × 6.25) was determined by Kjeldahl’s method [48] (method 984.13). The CP soluble fraction (expressed on a DM basis) was determined according to [49]. The neutral detergent (ND), acid detergent (AD) and lignin sulfuric acid (ADL) fiber fractions were sequentially determined using an AnkomII Fiber Analyzer (Ankom Technology Corporation, Fairport, NY, USA), as described by [50]. The ND solution contained sodium sulfite and a heat-stable amylase (activity: 17,400 Liquefon-U/mL, Ankom Technology Corporation, Fairport, NY, USA). The fiber fractions were corrected for residual ash (aND-Fom, ADFom). The starch content was determined by polarimetry (Polax 2 L, Atago^®^, Tokyo, Japan).

The mycotoxins were analyzed in cornmeal and TMR, with the samples examined before extraction. Mycotoxin detection/evaluation followed the method of [51] for aflatoxins (AFs); [52] for FBs; [53] for ZEN, DON, T-2 and HT-2 mycotoxins; and [54] for OTA. In summary, the extraction of AFB_1_ was performed using a ratio of acetone:water (7:3 *v*/*v*), after which it was purified by immuno-affinity column (R-Biopharm Rhône LTD, Glasgow, Scotland, United Kingdom). High-performance liquid chromatography (HPLC) with a fluorescence detector (FLD) was used to screen AFB_1_, with a limit of detection (LOD) and limit of quantification (LOQ) of 0.05 and 0.15 µg/kg DM, respectively. The FBs were extracted using a phosphate buffer and then purified using an immuno-affinity column (R-Biopharm Rhône LTD, Glasgow, Scotland, United Kingdom), after which quantification was completed by HPLC coupled with a mass spectrometer (HPLC–MS/MS) with an LOD and LOQ of 10 and 30 µg/kg DM, respectively. Extraction of the other mycotoxins, ZEN, DON, T-2 and HT-2 toxin, was achieved with acetonitrile:water (86:14 *v*/*v*). ZEN was purified using an immune-affinity column (R-Biopharm, Rhône LTD, Glasgow, Scotland, United Kingdom) and quantified using HPLC–FLD with LOD and LOQ values of 2 and 5 µg/kg DM, respectively. DON was purified using a Trilogy-puritox Trichothecenes column (R-Biopharm, Rhône LTD, Glasgow, Scotland, United Kingdom), followed by quantification using GC-MS, with LOD and LOQ values of 10 and 30 µg/kg DM, respectively. T-2 and HT-2 toxins were purified using a Trilogy-Puritox Trichothecenes column, followed by quantification by LC-MS/MS, with LOD and LOQ values of 0.5 and 1.5 µg/kg DM, respectively. 

### 5.3. Rumination Time, Body Weight and Body Condition Score 

The trial cows’ daily rumination time was detected using an accelerometer collar (RuminAct, SCR Heatime, Netanya, Israel), and the average times are reported. The cows were weighed twice daily using an electronic scale (TDM, San Paolo, Brescia, Italy) and the averages are reported. The metabolic BW was calculated using this formula: BW^0.75^ [29]. The BCS was determined at the start and the end of the experimental periods using the 5-point scoring system [55].

### 5.4. Health Status of Cows

The cows’ health status was monitored daily. Mastitis was diagnosed by visual evaluation (abnormal milk per quarter), routine milk conductivity level and somatic cell count (SCC) analysis, which was performed for suspicious cases. Diarrhea was diagnosed using the fecal score method (visual evaluation of consistency and color) [56]; diarrheic feces are those with a fecal score of 2 or less. A veterinarian visited all the animals, and no symptoms of disease were found before the start of the experimental periods. The husbandry was generally good, and the feces had normal consistency. Cows with severe diseases were not included in the trial. One animal fed the MMP diet was removed from the trial due to a health problem adjudged as non-conforming by the veterinarian. This animal was comprehensively treated for its symptoms but was excluded from the trial due to reduced milk production and worsening of health. Other animals had some minor and transient health problems that did not cause their exclusion from the experiment. These events were recorded. Consequently, some data related to specific animals with minor health injuries were excluded from the statistical analysis due to reasons independent of the treatments and related to other management problems. The excluded data represented less than 2.0% of all the data. 

### 5.5. Milk Yield, Composition and Cheesemaking Traits

The individual milk yield was measured at each milking, twice per day. Each week of the experiment period, representative daily milk samples were taken. The milk samples were analyzed for fat, protein, casein, lactose and titratable acidity using near-infrared spectroscopy (NIRs) (MilkoScan FT 120, Foss Electric, Hillerød, Denmark). The daily production of fat, protein, casein and lactose was calculated according to [57]. Urea nitrogen was determined in skimmed milk using a spectrophotometric assay and a urea nitrogen kit (cat# 0018255440, Instrumentation Laboratory, Milano, Italy) in association with an autoanalyzer (ILAB-650, Instrumentation Laboratory, Lexington, MA, USA). The somatic cell count (SCC) was determined using an optical fluorometric method with an automated cell counter (Fossomatic 180, Foss Electric, Hillerød, Denmark). In addition, bulk milk samples were analyzed weekly for AFM1, as described previously. The analyzed levels of AFM1 were under the LOD in all the samples. The milk coagulation properties (MCP) were evaluated using two mechanical lactodynamographs (Formagraph, Foss Electric A/S, Hillerød, Denmark). The following traditional MCP traits were recorded: rennet coagulation time (RCT, min), the time taken for the start of the coagulation after rennet addition; curd firming time (k20, min), the time to reach a curd firmness of 20 mm; and curd firmness (a30, mm) at 30 min after rennet addition. The instrument recorded the width of coagulation every 15 s for 60 min, resulting in a total of 240 measurements. The collection of measurements was used in the modeling equation proposed by [30] as an estimation of the following curd firming and syneresis traits: RCTeq (min) is the RCT estimated with the curd firming equation; k_CF_ (%min) is the curd firming rate constant; k_SR_ (%min) is the curd syneresis instant rate constant; CF_max_ (mm) is the maximum curd firmness reached within 45 min; and t_max_ is the time taken to reach CFmax. 

The cheese yield and milk nutrient recovery traits were analyzed in duplicate using the 9-Milca method [58]. Briefly, the cheesemaking procedure was mimicked by adding 200 µL of a rennet solution (Hansen Standard 215 with 80 ± 5% chymosin and 20 ± 5% pepsin; Pacovis Amrein AG) diluted to 1.2% (*w*/*v*) with fresh distilled water in 9 mL of milk. The initial incubation step was 30 min at 35 °C, after which a stainless-steel spatula was used to make the first cut. Next, a curd-cooking phase of 30 min at 55 °C was performed, during which a second manual cut was made. Once the cooking phase was finished, the curd was separated from the whey for 30 min at a temperature near to 25 °C, by gently applying pressure to the curd to better drain the whey. Lastly, the whey composition (fat, protein, lactose and TS) was measured with an FT2 infrared spectrophotometer (Milkoscan FT2; Foss Electric A/S, Hillerød, Denmark). This procedure resulted in seven cheesemaking traits. The curd weight (CY_CURD_), curd dry matter (CY_SOLIDS_) and water retained in the curd (CY_WATER_) are expressed as a percentage of the total milk processed. The curd nutrient recoveries (REC), namely, REC_PROTEIN_, REC_FAT_, REC_ENERGY_ and REC_SOLIDS_ (%), were calculated by measuring the difference in weight and composition between the milk and whey, according to [59]. For all the procedures, repeated measures were performed that were averaged prior to the statistical analysis. 

### 5.6. Feces Collection and Nutrient Digestibility

Fecal samples were collected from the rectum on days 0, 14, 28, 42 and 54 of the exposure periods. The feces were mixed, and subsamples were taken. The first subsample was dried at 60 °C, as described above. The ADL in this first subsample was used as an internal marker to calculate the digestion coefficients of the fecal samples.

### 5.7. Blood Sampling and Blood Biochemistry

Blood samples were collected for chemical analysis before start of the trial and after 1, 4 and 8 weeks of the experimental period. The samples were collected in the morning (before feeding) by venipuncture of the jugular vein in 10 mL Li–heparin-treated tubes (Vacuette, containing 18 IU of Li–heparin/mL, Kremsmünster, Austria). After collection, the samples were cooled immediately in an ice water bath. Part of the blood was used to calculate the packed cell volume (PCV) (Centrifugette 4203; ALC International Srl, Cologno Monzese, Italy). The other part was centrifuged (3500× *g* for 16 min at 4 °C) to collect the plasma, which was stored in aliquots at −20 °C until the analysis. 

The plasma metabolites were analyzed at 37 °C using an automated clinical analyzer (ILAB 650, Instrumentation Laboratory, Lexington, MA, USA), as described by [60]. Commercial kits from Instrumentation Laboratory SpA (Werfen, Italy) were used to measure the glucose, total cholesterol, urea, Ca, P, Mg, total protein, albumin, total bilirubin and creatinine. Kits from Wako (Chemicals GmbH, Neuss, Germany) were used to measure non-esterified fatty acids (NEFA), beta-hydroxybutyric acid (BHBA) and Zn. Electrolytes (Na, K and Cl) were measured using a potentiometer method (ion-selective electrode connected to ILAB 650). A kinetic analysis was used to determine the alkaline phosphatase (AP; EC 3.1.3.1), aspartate aminotransferase (AST; EC 2.6.1.1) and γ-glutamyltransferase (GGT; EC 2.3.2.2) activities, with kits from Instrumentation Laboratory SpA. The ceruloplasmin, haptoglobin, paraoxonase (PON) activity, myeloperoxidase (MPO) activity, reactive oxygen metabolites (ROMt) and ferric-reducing antioxidant power (FRAP) were measured as described by [61].

### 5.8. Statistical Analysis

Data were collected daily (i.e., dry matter intake or DMI, milk yield or MY, rumination time or RT and body weight or BW), weekly (milk composition and fat- or energy-corrected milk) or at a lower frequency during several occasions during the study (blood samples at the start and weeks 1, 4, and 8). Before the statistical analysis, all the data were tested for normality. Variables that tested as non-normal distributions (such as SCC) were log 10-transformed before the statistical analysis. The data were analyzed using the MIXED procedure in SAS, and for all the data that was measured more than once in each cow, analyses with repeated measures were included [62] according to the following model: Y_ijklm_ = μ + T_i_ + D_j_ + (T × D)_ij_ + p_k_ + c_l_ + e_ijklm_,
where Y_ijklm_ is the dependent variable, μ is the population mean, T_i_ is the fixed effect of the treatment, D_j_ is the fixed effect of the time (day or week) of measurement (repeated measurements), T × D_ij_ is the fixed effect of the treatment × time (day or week) of measurement interaction, pm is the fixed effect of the period, cl is the random effect of a cow and e_ijklm_ is the residual error. The same model was separately run for both adaptation and exposure periods. 

Five covariate model structures were used based on the finite sample-corrected Akaike information criterion (AICC) and the Schwarz Bayesian criterion for the best fitting model. The five tested structures were compound symmetry, heterogeneous compound symmetry, unstructured, auto-regressive (1) and anti-dependence [62,63]. A *p* value below 0.05 was considered significant, and a *p* value between 0.05 and 0.10 was considered to indicate a trend. In the tables, *p* values lower than 0.001 are reported as <0.001. If the main effects (i.e., T and W) or the first order interaction (T * W) were significant, a post-hoc multiple comparison T test was adopted to verify the differences among the least square means.

## Figures and Tables

**Table 1 toxins-15-00546-t001:** Chemical composition, digestibility and energy evaluations of experimental total mixed ration (TMR) diets fed to lactating dairy cows in the trial: CTR diet with low contamination level, MTX with high contamination level and MMP with high contamination level supplemented with about 100/g/cow/day of MMP (i.e., TOXO^®^ HP-R, Selko, Tilburg, The Netherlands).

Items	Experimental Diets ^1^		
	CTR (*n* = 10)	MTX (*n* = 10)	MMP (*n* = 10)
Ingredients (% DM)			
Corn meal		5.9	
Barley meal		2.5	
Sunflower meal, dehulled 34%		4.3	
Soybean, solvent meal 44%		13.7	
High moisture corn		22.7	
Alfalfa hay		17.4	
Ryegrass hay		1.8	
Mineral-vitamin supplement ^2^		1.7	
Fat (palm oil)		0.8	
Corn silage		12.0	
Sorghum silage		12.2	
Beet pulp		5.0	
Forage:concentrate ratio		49.9:50.1	
Chemical composition (% DM)			
DM (% as fed)	53.8 ± 2.4	52.7 ± 2.3	53.6 ± 1.9
CP	14.8 ± 1.0	14.6 ± 1.0	15.1 ± 0.8
Soluble CP	5.2 ± 0.5	5.2 ± 0.6	5.3 ± 0.5
Ash	8.9 ± 0.6	8.8 ± 0.6	8.7 ± 0.6
ANDFom	31.9 ± 1.8	31.9 ± 2.4	32.0 ± 1.2
ADFom	19.7 ± 1.4	20.3 ± 1.8	19.7 ± 1.8
ADL	3.0 ± 0.3	3.1 ± 0.4	2.9 ± 0.3
NDFD 24 h	47.9 ± 2.1	48.3 ± 2.3	48.3 ± 1.9
EE	3.3 ± 0.5	3.3 ± 0.5	3.3 ± 0.4
Starch	26.9 ± 1.7	26.8 ± 1.9	27.1 ± 1.9
Sugar	4.2 ± 0.5	4.2 ± 0.4	4.3 ± 0.6
NDICP	3.1 ± 0.5	3.0 ± 0.5	3.1 ± 0.5
ADICP	0.9 ± 0.1	0.9 ± 0.1	0.9 ± 0.1
Energy evaluations (Mcal/kg DM) ^3^			
TDN (%)	70.1 ± 0.7	70.0 ± 0.9	70.1 ± 0.8
ME3x	2.54 ± 0.05	2.54 ± 0.06	2.54 ± 0.05
Mycotoxin contamination ^4^ in TMR (µg/kg DM)			
FB1 in TMR	85.3 ± 56.3	159.5 ± 60.9	163.8 ± 58.1
FB2 in TMR	44.3 ± 30.4	75.9 ± 31.9	77.9 ± 32.9
ZEN in TMR	43.2 ± 13.1	196.8 ± 75.7	248.5 ± 139.3
DON in TMR	284.9 ± 91.9	1021.7 ± 234.5	1009.6 ± 213.5

^1^ Treatments consisted of (i) the CTR diet, TMR with low contamination HMC and beet pulp; (ii) the MTX diet, TRM with highly contaminated HMC and beet pulp; (iii) the MMP diet, MTX diet supplemented with 100 g/cow/day of mycotoxin mitigation product (MMP, Selko, Tilburg, The Netherlands). ^2^ Mineral–vitamin supplement composition: sodium bicarbonate; 900,000 IU of vitamin A; 150,000 IU of vitamin D3; 3000 mg of vitamin E; 2000 mg encapsulated niacinamide; 20,000 mg niacinamide; 20,000 mg choline chloride; 1100 mg of copper(I) sulfate; 1300 mg of MgO; 9400 mg of zinc sulfate; 65 mg of potassium iodide; 30 mg of sodium selenite; 18,000 mg DL-methionine. ^3^ Energy evaluations were calculated using equations of [29]. ^4^ Aflatoxins and T2 and HT2 toxins were undetected in all samples.

**Table 2 toxins-15-00546-t002:** Chemical and fermentative (only silage) characteristics of single feeds (*n* = 4). Nd—not detected. Aflatoxins and T2 and HT2 toxins were undetected in all samples.

	Feed Ingredients							
Chemical Composition (% DM)	Corn Silage	Sorghum Silage	HMC CTR	HMC TRT	Beet Pulp TRT	Beet Pulp CTR	Alfalfa Hay	Ryegrass Hay
DM, % as fed	32.68 ± 1.9	29.4 ± 0.1	54.7 ± 1.4	58.13 ± 0.4	89.1 ± 1.3	88.8 ± 1.1	89.0 ± 0.4	90.0 ± 0.3
CP	8.4 ± 0.7	10.6 ± 0.3	5.9 ± 0.1	6.3 ± 0.3	8.1 ± 0.7	8.9 ± 0.6	15.0 ± 3.4	6.2 ± 0.7
Ash	5.7 ± 0.5	7.7 ± 0.5	1.2 ± 0.1	1.1 ± 0.2	6.8 ± 1.3	7.1 ± 1.5	10.0 ± 1.2	9.8 ± 1.0
aNDFom	40.0 ± 2.8	48.3 ± 0.3	16.4 ± 0.5	17.8 ± 2.3	40.5 ± 4.9	44.1 ± 3.7	44.9 ± 7.6	59.2 ± 1.0
ADFom	25.2 ± 1.9	31.7 ± 0.7	9.4 ± 0.1	8.4 ± 2.1	20.6 ± 2.0	23.8 ± 1.5	33.9 ± 2.0	40.7 ± 1.6
ADL	3.2 ± 0.7	4.6 ± 0.1	-	-	1.9 ± 0.8	1.7 ± 0.7	7.2 ± 0.7	6.0 ± 0.4
NDFD_24h_, %NDF	50.9 ± 7.2	43.6 ± 0.6	-	-	-	-	32.8 ± 6.4	45.9 ± 2.6
EE	3.2 ± 0.3	3.6 ± 0.1	-	-	0.9 ± 0.4	0.9 ± 0.4	1.9 ± 0.5	1.3 ± 0.3
Starch	21.3 ± 0.8	20.6 ± 1.1	59.6 ± 2.7	60.3 ± 4.1	-	.	2.9 ± 0.2	3.5 ± 0.8
Sugar	1.3 ± 0.5	2.1 ± 0.1	-	-	6.6 ± 2.2	7.1 ± 2.4	7.9 ± 1.5	9.2 ± 0.2
Mycotoxins (µg/kg DM)								
FB1	96.2 ± 8.7	106.2 ± 13.2	<10	645.3 ± 43.2	<10	13.1 ± 2.4	<10	<10
FB2	38.3 ± 4.8	21.4 ± 1.3	<10	278.3 ± 25.9	<10	<10	<10	<10
ZEN	<10	238.2 ± 21.9	<10	76.1 ± 21.1	2911.4 ± 121.4	<10	<10	<10
DON	713.2 ± 54.2	635.6 ± 47.2	38.3 ± 13.1	1988.1 ± 234.2	45.3 ± 5.6	27.5 ± 3.8	519.2 ± 78.5	355.3 ± 43.2
Fermentation parameters (%DM)								
pH	3.68 ± 0.05	3.83 ± 0.02						
Ethanol	0.52 ± 0.34	1.10 ± 0.13						
Acetic acid	3.02 ± 0.55	3.55 ± 0.63						
Propionic acid	0.06 ± 0.02	0.28 ± 0.15						
Isobutyric acid	0.00 ± 0.00	0.00 ± 0.00						
1,2 Propanediol	2.42 ± 0.97	1.87 ± 1.12						
Butyric acid	0.02 ± 0.02	0.01 ± 0.01						
Isovalerianic acid	0.00 ± 0.00	0.00 ± 0.00						
Valerianic acid	0.00 ± 0.00	0.00 ± 0.00						
Lactic acid	5.77 ± 0.91	6.40 ± 0.49						
Aldehydes tot.	0.01 ± 0.00	0.01 ± 0.00						
Alcohols tot.	0.75 ± 0.47	1.87 ± 0.41						
Ketons tot.	0.00 ± 0.00	0.00 ± 0.00						
Esters tot.	0.02 ± 0.01	0.04 ± 0.01						

**Table 3 toxins-15-00546-t003:** Least squares means and associated SEM for feeding behavior, body weight, milk yields, feed efficiency and milk parameters of 36 Holstein cows (12 for each experimental group) fed with 3 experimental diets: CTR diet with low contamination level, MTX with high contamination level and MMP with high contamination level supplemented with about 100/g/cow/day of MMP (i.e., TOXO^®^ HP-R, Selko, Tilburg, The Netherlands) during the adaptation period. Periods 1 and 2 refer, respectively, to March/May and May/July 2022.

Items		Treatment			Period		SEM	P of the Model			
		CTR	MTX	MMP	1	2		Period	Treatment (T)	Day (D)	D * T
**Feeding Behavior**											
DMI	kg/cow/day	25.21	25.18	25.67	24.34	26.36	1.270	0.032	0.896	0.132	0.328
DMI	% BW	4.01	3.85	4.03	3.81	4.17	0.031	0.046	0.503	0.140	0.247
Rumination time	Min	519	504	526	519	513	50.2	0.848	0.833	0.022	0.328
**Body weight**	Kg	622	653	638	641	635	17.6	0.651	0.080	0.016	0.284
**Milk yields**											
Milk yield	L/cow/day	36.9	36.7	35.8	35.6	37.4	1.08	0.465	0.928	<0.001	0.621
Milk yield	kg/cow/day	38.0	37.8	36.8	36.7	38.4	1.15	0.465	0.982	<0.001	0.621
FPCM	kg/cow/day	38.5	37.7	38.6	34.4	40.7	6.70	0.060	0.637	-	-
ECM	kg/cow/day	35.5	34.2	35.8	33.2	37.7	6.16	0.051	0.653	-	-
Milk yield/DMI	dmnl	1.53	1.53	1.46	1.53	1.48	0.010	0.531	0.763	0.168	0.539
**Milk parameters**											
Fat	%	3.71	3.91	3.99	4.17	3.56	0.406	<0.001	0.522	-	-
	kg/cow/day	1.37	1.30	1.42	1.30	1.42	0.270	0.387	0.667	-	-
Protein	%	3.23	3.28	3.28	3.29	3.22	0.235	0.578	0.927	-	-
	kg/cow/day	1.20	1.10	1.17	1.02	1.28	0.199	0.012	0.618	-	-
Casein	%	2.55	2.60	2.61	2.62	2.55	0.279	0.613	0.924	-	-
	kg/cow/day	0.94	0.87	0.93	0.81	1.01	0.160	0.016	0.624	-	-
Lactose	%	4.79	4.84	4.83	4.79	4.85	0.166	0.359	0.785	-	-
	kg/cow/day	1.78	1.64	1.74	1.49	1.94	0.329	<0.001	0.703	-	-
MUN	mg/100 mL	25.3	27.5	24.7	29.5	22.1	4.30	<0.001	0.552		
LogSCC	Log10(cells/mL)	5.01	5.03	4.67	5.21	4.60	0.668	0.065	0.540	-	-

dmnl = dimensionless.

**Table 4 toxins-15-00546-t004:** Least squares means and associated SEM for feeding behavior, body weight, milk yields, feed efficiency and milk parameters of 36 Holstein cows (12 for each group) fed with 3 experimental diets: CTR diet at low contamination level, MTX at high contamination level and MMP at high contamination level supplemented with about 100/g/cow/day of MMP (i.e., TOXO^®^ HP-R, Selko, Tilburg, The Netherlands) during each experimental exposure period. Periods 1 and 2 refer, respectively, to March/May and May/July 2022.

Items		Treatment			Period		SEM	P of the Model			
		CTR	MTX	MMP	1	2		Period	Treatment (T)	Day (D)	D * T
**Feeding Behavior**											
DMI	kg/cow/day	25.62	26.09	26.36	26.47	25.58	0.401	0.238	0.714	<0.001	0.987
DMI	% BW	4.02	3.98	4.09	4.10	3.96	0.01	0.290	0.829	<0.001	0.986
Rumination time	min	512	505	524	518	510	14.53	0.684	0.756	<0.001	0.800
**Body weight**	kg	638	657	647	646	648	8.0	0.858	0.358	<0.001	0.835
Body condition score	1–5 scale	3.17	3.24	3.18	3.10	3.28	0.023	0.212	0.456	-	-
**Milk yields**											
Milk yield	L/cow/day	37.2	36.5	38.2	37.6	37.1	0.27	0.834	0.839	<0.001	0.959
Milk yield	kg/cow/day	38.3	37.6	39.4	38.7	37.2	0.29	0.834	0.839	<0.001	0.959
3.5% FCM	kg/cow/day	39.9	39.7	41.6	41.3	39.5	0.762	0.431	0.752	0.008	0.017
ECM	kg/cow/day	37.8	36.6	38.4	38.1	36.5	0.669	0.436	0.761	0.006	0.02
Milk yield/DMI	dmnl	1.54	1.46	1.52	1.50	1.51	0.076	0.895	0.692	0.02	0.999
**Milk parameters**											
Fat	%	3.52	3.74	3.80	3.77	3.60	0.011	0.133	0.138	<0.001	0.565
	kg/cow/day	1.37	1.42	1.49	1.47	1.38	0.002	0.245	0.374	<0.001	0.212
Protein	%	3.36	3.26	3.31	3.31	3.30	0.016	0.921	0.383	0.007	0.194
	kg/cow/day	1.31	1.23	1.30	1.30	1.26	0.0007	0.643	0.930	0.223	0.006
Casein	%	2.67	2.59	2.63	2.63	2.63	0.0018	0.993	0.775	0.003	0.274
	kg/cow/day	1.04	0.98	1.03	1.03	1.01	0.0005	0.689	0.907	0.024	0.365
Lactose	%	4.75	4.74	4.80	4.75	4.77	0.009	0.639	0.751	0.017	0.939
	kg/cow/day	1.86	1.81	1.90	1.87	1.84	0.0013	0.741	0.910	0.276	0.030
MUN	mg/100 ml	31.6	32.7	33.7	31.9	33.5	0.859	0.125	0.301	0.002	0.965
LogSCC	Log10(cells/mL)	4.68	4.88	4.60	4.81	4.63	0.006	0.400	0.506	0.119	0.142

dmnl = dimensionless.

**Table 5 toxins-15-00546-t005:** Least squares means and associated SEM for nutrient diet digestibility and volatilome of feces of 36 Holstein cows (12 for each group) fed with 3 experimental diets: CTR diet at low contamination level, MTX at high contamination level and MMP at high contamination level supplemented with about 100/g/cow/day of MMP (i.e., TOXO^®^ HP-R, Selko, Tilburg, The Netherlands) during each experimental exposure period. Periods 1 and 2 refer, respectively, to March/May and May/July 2022.

Items		Treatment			Period		SEM	P of the Model			
		CTR	MTX	MMP	1	2		Period	Treatment (T)	Week (W)	W * T
**Diet digestibility**											
Apparent NDF digestibility	%	65.68	66.87	64.27	66.35	64.85	0.632	0.039	0.126	0.021	0.963
Apparent starch digestibility	%	98.21	98.12	97.48	98.30	97.57	0.050	<0.001	0.060	0.610	0.974
Apparent CP digestibility	%	81.61	81.84	79.32	81.73	80.11	0.287	<0.001	<0.001	0.363	0.612
**Fecal fermentation profile**											
pH of feces	dmnl	6.83	6.86	6.76	6.96	6.68	0.002	<0.001	0.378	0.504	0.919
**Volatilome fecal profile**											
Acetic acid	mmol/kg DM	167.63	169.25	178.99	148.73	195.17	168.13	<0.001	0.683	0.04	0.513
Propionic acid	mmol/kg DM	47.39	48.85	48.27	40.31	56.03	12.201	<0.001	0.946	0.02	0.998
Butyric acid	mmol/kg DM	22.02	22.52	24.73	19.00	27.17	3.523	<0.001	0.595	0.004	0.556
Isobutyric acid	mmol/kg DM	2.34	2.95	2.48	2.27	2.91	0.073	<0.001	0.142	0.058	0.922
Isovalerianic acid	mmol/kg DM	2.14	2.71	2.20	1.81	2.89	0.079	<0.001	0.066	<0.001	0.578
Valerianic acid	mmol/kg DM	3.39	3.69	3.47	2.96	4.06	0.096	<0.001	0.393	<0.001	0.384
Methanol	mmol/kg DM	0.76	0.29	0.28	0.32	0.56	0.200	0.288	0.296	0.438	0.249
Ethanol	mmol/kg DM	4.83	4.01	5.81	4.22	5.54	1.906	0.086	0.696	0.476	0.836
Volatile fatty acid tot.	mmol/kg DM	244.90	249.96	260.13	215.09	288.24	320.11	<0.001	0.836	0.002	0.634
Aldehydes tot.	mmol/kg DM	2.51	1.85	2.33	2.60	1.87	0.528	0.070	0.392	0.002	0.740
Alcohols tot.	mmol/kg DM	6.41	4.78	6.88	5.14	6.92	2.883	0.040	0.585	0.913	0.801

Dmnl, dimensionless. Capronic acid, propylene glycol, acetone, ketones and esters were analyzed but not detected.

**Table 6 toxins-15-00546-t006:** Least squares means and associated SEM for blood parameters of 36 Holstein cows (12 for each group) fed with 3 experimental diets: CTR diet at low contamination level, MTX at high contamination level and MMP at high contamination level supplemented with about 100/g/cow/day of MMP (i.e., TOXO^®^ HP-R, Selko, Tilburg, The Netherlands) during each experimental exposure period. Periods 1 and 2 refer, respectively, to March/May and May/July 2022.

Items		Treatment			Period		SEM	P of the Model			
		CTR	MTX	MMP	1	2		Period	Treatment (T)	Week (W)	W * T
Plasma components											
Indexes of energy metabolism–protein metabolism											
Glucose	mmol/L	4.36	4.28	4.28	4.31	4.30	0.008	0.895	0.819	0.853	0.704
Cholesterol	mmol/L	4.74	4.81	4.61	4.55	4.89	0.059	0.190	0.576	0.210	0.624
NEFA	mmol/L	0.10	0.10	0.13	0.11	0.10	0.007	0.734	0.245	0.009	0.549
BOHB	mmol/L	0.31	0.38	0.36	0.37	0.34	0.003	0.450	0.051	0.148	0.074
Urea	mmol/L	5.56	6.18	5.39	5.93	5.49	0.158	0.244	0.712	0.032	0.751
Creatinine	μmol/L	84.45	82.59	82.49	80.41	85.95	1.679	<0.001	0.814	<0.001	0.202
Indexes of mineral metabolism											
Calcium	mmol/L	2.52	2.48	2.44	2.46	2.50	0.040	0.235	0.290	0.871	0.404
Phosphorous	mmol/L	1.71	1.86	1.73	1.82	1.72	0.018	0.268	0.764	0.524	0.429
Magnesium	mmol/L	1.08	1.05	1.06	1.03	1.10	0.020	0.010	0.972	<0.001	0.622
Zinc	mcmol/L	16.47	17.07	16.74	16.18	17.34	0.918	0.311	0.988	0.159	0.946
Indexes of liver functionality											
GGT	U/L	33.28	27.64	30.05	27.74	32.91	3.288	0.052	0.317	0.016	0.420
GOT	U/L	122.50	102.25	109.64	105.63	117.30	8.080	0.311	0.250	0.277	0.382
Alkaline phosphatase	U/L	51.18	57.81	61.52	57.92	55.84	26.898	0.775	0.515	0.652	0.034
Albumin	g/L	36.07	35.93	35.83	35.47	36.42	0.345	0.049	0.377	0.019	0.149
Bilirubin	mcmol/L	1.65	1.53	1.49	1.45	1.66	0.021	0.105	0.892	0.310	0.754
Paraoxonase	U/ml	90.48	85.40	90.57	90.04	87.59	17.277	0.518	0.245	0.782	0.098
Indexes of innate immune system and oxidative stress											
Haptoglobin	g/L	0.28	0.24	0.19	0.25	0.22	0.040	0.495	0.205	0.178	0.562
Ceruloplasmin	mcmol/L	2.93	2.77	2.70	2.80	2.80	0.027	0.972	0.602	0.698	0.613
Total proteins	g/L	83.19	84.18	82.49	83.52	83.05	1.672	0.669	0.270	0.181	0.342
Globulin	g/L	47.12	48.23	46.66	48.05	46.63	1.914	0.280	0.160	0.819	0.130
MPO	U/L	446.65	466.71	470.62	453.86	468.79	476.04	0.279	0.755	0.008	0.991
Total antioxidants (FRAP)	μmol/L	132.81	133.57	134.48	127.77	139.48	39.783	0.017	0.751	0.130	0.515
ROM	mgH_2_O_2_/100 mL	16.33	15.85	15.19	15.86	15.73	0.867	0.891	0.848	0.145	0.740

**Table 7 toxins-15-00546-t007:** Least squares means and associated SEM for milk coagulation properties of 36 Holstein cows (12 for each group) fed with 3 experimental diets: CTR diet at low contamination level, MTX at high contamination level and MMP at high contamination level supplemented with about 100/g/cow/day of MMP (i.e., TOXO^®^ HP-R, Selko, Tilburg, The Netherlands) during each experimental exposure period. Periods 1 and 2 refer, respectively, to March/May and May/July 2022.

Items		Treatment			Period		SEM	P of the model			
		CTR	MTX	MMP	1	2		Period	Treatment (T)	Week (W)	W * T
Milk coagulation properties											
pH	dmnl	6.46	6.49	6.47	6.51	6.44	0.008	<0.001	0.158	0.0023	0.533
Casein index	%	78.8	79.4	80.2	80.3	78.7	0.800	0.001	0.225	0.006	0.624
Milk total solid	%	12.69	12.76	12.87	13.01	12.54	0.04	0.108	0.666	0.009	0.074
Milk total solid (w/o fat)	%	9.21	9.11	9.15	9.24	9.08	0.007	0.196	0.662	0.049	0.443
r	Min	24.2	29.0	25.8	26.9	25.8	6.92	0.744	0.171	0.418	0.274
K20	Min	8.4	11.9	8.7	10.0	9.3	1.65	0.719	0.237	0.225	0.062
a30	Mm	17.9	13.2	23.7	18.1	18.5	11.98	0.927	0.338	0.313	0.460
a45	Mm	29.3	24.7	32.3	31.0	26.4	15.64	0.251	0.177	0.948	0.202
a60	Mm	29.0	26.3	28.6	31.6	24.4	28.58	0.024	0.408	0.486	0.514
Rct_eq	Min	24.9	29.0	21.3	25.5	24.6	3.69	0.716	0.025	0.781	0.112
tmax	Min	48.0	52.3	43.1	50.8	44.9	10.15	0.051	0.085	0.545	0.847
CFmax	Mm	34.3	28.5	35.5	35.7	29.8	12.39	0.074	0.079	0.957	0.082
CFp	Mm	46.0	38.2	47.5	47.9	40.0	22.25	0.074	0.079	0.957	0.082
kcf	%/min	10.3	9.2	12.7	9.5	11.9	4.17	0.081	0.793	0.216	0.544
ksr	%/min	0.9	0.8	1.3	0.8	1.2	0.09	0.085	0.829	0.077	0.410
CY parameters											
CY curd	%	18.91	18.51	19.62	20.23	17.79	1.089	0.002	0.924	0.030	0.703
CY solid	%	6.51	6.30	6.59	6.70	6.24	0.074	0.163	0.997	0.010	0.303
CY water	%	12.41	11.95	13.02	13.53	11.38	1.010	<0.001	0.577	0.201	0.958
REC parameters											
REC protein	%	77.82	76.12	79.32	78.60	76.91	2.480	0.148	0.272	0.207	0.923
REC fat	%	82.78	80.65	83.54	82.42	82.22	16.124	0.951	0.575	0.078	0.363
REC solids	%	51.20	48.88	51.06	51.31	49.44	3.693	0.266	0.844	0.135	0.698
REC energy	%	64.75	63.04	65.63	65.18	63.77	2.835	0.470	0.855	0.017	0.457

r = coagulation time; K20 = time interval between gelation and attainment of curd firmness of 20 mm; a30, a45 and a60 = curd firmness 30, 45 and 60 min after rennet addition; RCTeq = rennet coagulation time estimated by curd firming equation parameter modeling; tmax = time at achievement of maximum curd firmness (CFmax); CFP = asymptotic potential curd firmness; CFmax = maximum curd firmness; kCF = curd firming instant rate constant; kSR = syneresis instant rate constant; CY = cheese yields; CY curd = cheese yields curd; CY solid = cheese yields for total solids; CY water = cheese yields for water; REC protein = recovery of protein; REC fat = recovery of fat; REC solids = recovery of solids; REC energy = recovery of energy.

## Data Availability

Data supporting the reported results, including links to publicly archived datasets analyzed or generated during the study, are available on request from the corresponding author.

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
