# Peer review of "Effects of Supplementation of a Mycotoxin Mitigation Feed Additive in Lactating Dairy Cows Fed Fusarium Mycotoxin-Contaminated Diet for an Extended Period"

_toxins, 2023, doi:10.3390/toxins15090546_

Round 1
Reviewer 1 Report
This manuscript summarizes a study conducted with dairy cows to evaluate the mycotoxin mitigation effects of a commercial product. Three groups of dairy cows (n=6/group) were evaluated during two periods lasting about 54 days each: control cows, cows fed contaminated feed, cows fed contaminated feed plus the mitigation product. The study was comprehensive in the milk component analysis, cow blood parameters measured and cheese related measurements of milk. Most results were not significant, but a few were significant. ECM and a cheese clotting measure were reduced in cows fed mycotoxin contaminated feed and the mitigation product prevented these reductions. Also, plasma BHOB was increased in mycotoxin fed cows and this increase was partially reduced feeding the mitigation product. A control group fed the mitigation product only without mycotoxins would have been ideal. A similar study with a different mitigation product was previously published by the authors (Gallo et al., 2020) and could be better discussed relative to the present study.
Major comments:
11. These findings are valuable, but the authors need to re-write their results and discussion not discussing “numerical differences”. Statistical results should be adhered to and not minimized by discussing “numerical differences”. This is especially true for line 443-461. It should also be stated that perhaps a study with a larger sample size may be needed. The amounts of mycotoxin fed and number of animals per group is not clearly discussed as a potential cause for differences among studies.
22. Also, several key variables measured had significant Day x treatment or Week x treatment effect, yet none of these interactions were described or discussed. It would have been interesting to see how the variables changed over time with the various treatments, especially with ECM.
33. Reference numbers cited in text are not correct and some are duplicated; authors need to verify each citation. For examples, reference 2 and 8 are the same; Line 242 ref. 40 is not correct; line 257, ref. 43 is not correct.
44. Line 389-397: Here and elsewhere, accurate reiteration of results is needed here, using significant statistical results. Treatment and Week x Treatment effects were not significant for cholesterol, urea or MPO (all P>0.19). Please reword.
Additional comments:
11. Line 59-60: this statement should have a reference cited.
22. Line 75-78: this statement should have a reference cited.
33. Line 104: If possible, milking times should be listed.
44. Line 205: should this be “properties” not “propertied”? Also, should state “…carred out in….” not “…duplicate out….”
55. Line 223: list “room temperature” in degrees C (or range of C).
66. Line 337-338: please define/spell out what FPCM and ECM are referring to.
77. Line 367: should this be “nutrients”?
88. Line 391-393: here and elsewhere, please correct use of abbreviations: what is correct “MXT” or “MTX”?
99. Line 482: Delete “different” as it was stated in previous sentence “there were no significant differences”.
110. Line 508: doesn’t table 7 show a significant treatment effect on Rct_eq?
111. Line 522-523: Neither Treatment or Week x Treatment effects were significant for GOT or GGT, so this statement is not correct.
112. Line 532: use “ECM” rather than “milk yield”.
113. Line 538: add “…should be carried out with a larger number of cows to verify….”
A few grammatical errors need to be corrected.
Reviewer 2 Report
The authors of this manuscript aimed to determine the effects of a commercially available mycotoxin mitigation product in lactating dairy cows fed a Fusarium mycotoxin-contaminated diet. They evaluated the effects on feeding behavior, immuno-physiological parameters, milk yield, and quality, cheese-making traits, and the overall health status of multiparous cows. Although most of the results did not show significant differences among treatments, this study is considered one of the scarce and important applied studies in the field of studying the effect of mycotoxins in ruminants, especially dairy cows. The low dose of the feed additive or their ineffectiveness may be the reason for the absence of significant differences. Additionally, not using a control group without mycotoxin is considered one of the main weaknesses of the study. Here are some comments on the current manuscript:
1- Why was the control group without the use of mycotoxin not included in the study instead of considering the low-concentration group as a control?
2- Permissible limits in the dairy cow diets of the tested mycotoxins defined by the official authorities should be compared with the obtained results in Table 1.
3- Abstract: The results are concise and not representative of the study. Please add more information about the results obtained.
4- L25-28: The conclusion section should be rewritten, as in the current form, it is a concise sentence. The conclusion should answer the aim of the study.
5- The introduction section primarily well-covered the adverse effects of mycotoxins on ruminants, with a particular focus on production performance. However, it does not address why the tested commercial product or its components were chosen as a mitigation strategy for Fusarium mycotoxin impacts. Additionally, what are the advantages of the commercial formulation being studied compared to other methods of detoxification?
6- The hypothesis of the study should be clearly stated at the end of the Introduction section.
7- Were the treatments administered to the same cows during both the first and second periods, or were different cows used?
8- Can you provide more information on how the high moisture corn (HMC) with low or high contamination levels was prepared as a source of mycotoxins?
9- Many references are cited in the text while it is not in the list of references (e.g. lines 139, 145, 175....etc). Please review and revise this issue accordingly.
10- The instruments used (i.e., L206, L203, ……etc) must contain all full information (i.e., model, company, city, country).
11- In all Tables, describe the experimental groups and all abbreviations used in the table's footnotes. Describe also number of replicates per treatment (n=?). Replace “sem” with “SEM” throughout the text and all tables.
12- Why is the P value shown as <0.05 (e.g. pages 13 and 16)? It is unclear why the actual value is not shown instead.
-
Reviewer 3 Report
As a result of the review of the paper entitled „ Effects of supplementation of a mycotoxin mitigation feed additive in lactating dairy cows fed Fusarium mycotoxin-contaminated diet for an extended period”, a number of more or less detailed comments are presented below:
Line (L) 12 – Probably "quantitative share of mycotoxins";
L 31 – Perhaps 'state of health';
L 40 – I think "In all climatic zones. Where plant material is available';
L 41, 153, 434 – mycotoxins;
L 59 – As suggested by EFSA, the abbreviation ZEN should be used throughout the work;
L 96, 107, 271, 272, 353, 361, 378, 384, 386, 398 – This was not a treatment of cows but a detoxification of feed;
L 132, 274, 335, 343, 349, 368, 370 – not poisoning but exposure;
L 141 and 145 – literature misquoted;
L 153 – In my opinion, studies of mycotoxin metabolites are lacking. They are often found in feed materials because mycotoxin biotransformation processes already take place in the plant organism and are much more dangerous than the parent substances;
L 157, 160, 165 and 169 – after the units should be added dry matter;
L 183 – Perhaps not so much healthy, but rather in whom no symptoms of disease were found;
L 235 – There is no such vein. There is a zygomatic vein (Vena jugularis externa). This method has some drawbacks, because this blood is physiologically detoxified after passing through the liver system. More reliable from a diagnostic point of view is the collection of blood from Vena caudalis median, since it is a collection blood from the gastrointestinal tract but before the hepatic portal system;
L 482 – It would probably be better to call diets rather than treatment, because the feed was affected and not directly the cow;
L 487 – Instead of "mycotoxin-fed" it would be better to describe "exposed";
L 493 – not so much inverted (from what?) as slowed down or accelerated.
Reviewer 4 Report
Abstract:
The connection between the test steps and the results in the abstract is stiff, so it is suggested to add a connective word.In addition, the summary lacks a conclusion at the end and adds a concluding statement.
The experimental steps in the abstract do not mention the determination of cheese making properties, but the purpose of this study and the evaluation content mentioned in the last sentence of the last paragraph of the preface are both mentioned in the abstract. It is suggested to add.
Introduction
Each sentence in the first paragraph of Introduction is a reference to literature, similar to the content in the discussion. It is suggested to add your own sentence.
It is recommended to give examples of which mycotoxins alter rumen flora due to their antibacterial activity, as the above mentioned mycotoxins are all given examples to maintain consistency.
Materials and Methods section
1. Provide more details about the CERZOO research and experimental center where the study was conducted. Include information about its facilities, expertise, and previous research in the field.
2. Clarify the process of grouping cows based on days in milk (DIM), parity, milk yield, and body condition score. Explain the rationale behind this grouping and its relevance to the study.
3. Specify the exact procedures for the analysis of feeds, diets, and mycotoxins. Include the specific equipment used, the steps taken, and the methods for sample preparation and data analysis.
4. Provide more information about the health status monitoring of cows. Describe the criteria used for diagnosing mastitis, diarrhea, and other health issues. Include details about the veterinarian's assessment and decision-making process.
5. Include a clear explanation of the statistical analysis used, including the assumptions made and the choice of models. State the reasons for selecting specific model structures.
6. Consider providing a summary or table with key characteristics of the experimental groups, such as age, weight, and breed, to help readers understand the cow population under study.
7. Expand on the information about the mycotoxin mitigating product (TOXO® HP-R). Include its composition, mode of action, and previous studies supporting its effectiveness.
Results
The effect of this study on the feeding behavior of cows is mentioned in the abstract. In which experimental method?And other research shows that.
The conclusions are consistent with the recommendations in Table 1-4, but not in Table 5-7.
The detection of blood biochemistry is only mentioned in the report in Table 6. The plasma parameters are mentioned at the end, and it is suggested to supplement the content.
The test results should correspond to the test methods. For example, the milk yield in the results is mentioned in Table 3, while the test methods are mentioned after the measurement of rumination time, physical condition and weight.
Discussion
Discussion should be explained with periods totaling the total score, and the order of the discussion should also be consistent with material methods and results.
The first paragraph of 4.4 was discussed among the experimental groups, and it was speculated that the MMP group could make better or more efficient use of nutrients.It is not recommended to use vague words here, but to be justified.
In conclusion, it is suggested that the author put forward some suggestions on alleviating mycotoxins by feed additives and the prospect for the future.
Reviewer 5 Report
The paper, titled “Effects of supplementation of a mycotoxin mitigation feed additive in lactating dairy cows fed Fusarium mycotoxin-contaminated diet for an extended period”, evaluated a mycotoxin mitigation product (HP-R) in lactating dairy cows fed a fusarium-mycotoxin contaminated diet and the repercussions on feeding behavior, milk yield, milk quality, cheese-making traits and health status of cows. This manuscript was well-organized and well-debated. A few minor issues are listed below, however, this does not affect the publication of the manuscript in this journal.
1. Line 21-22: The abbreviations of CTR, TMR, C-B and MTX should be given the full name at their first mention.
2. Line 29: I do not think “blood” is a suitable keyword.
3. Line 152-170: I notice that the authors applied the LC-MS/MS method to determine FBs. Analytical methods of multi-mycotoxins by LC-MS/MS have been wildly adopted by laboratories in the EU, due to their high sensitivities and accuracies. So, why other mycotoxins were not detected using LC-MS/MS?
4. Line 203-204: Please indicate the cited literature for the detection method of AFM1.
5. Line 314-315: how many replicates for each test? The SD value of Zearalenone seems to be high.
6. Line 329: The abbreviations of SEM should be given the full name (standard error of the mean) at its first mention.
7. Line 344: the “sem” in the Table 3 should be “SEM”.
8. Line 372: I can not understand the meaning of the data in the columns Period 1 and Period 2 of Table 4. According to my understanding, each period should contain three sets of data (CTR, MTX, and MMP).
Round 2
Reviewer 2 Report
Dear Authors,
Thank you for the extensively revised version of the manuscript. On the basis of the improvement placed, the manuscript now is suitable for publication in the Toxins journal.
Best Regards
Reviewer 3 Report
No comments